# Intersectional disparities in mental healthcare utilization by sex and race/ethnicity among US adults: An NHANES study

Lotenna Olisaeloka[1,2]*, Esteban J. Valencia[1], Gentille Musengimana[1], Daniel Vigo[1,2], Ehsan Karim[1]

1 School of Population and Public Health, University of British Columbia, Vancouver, Canada,
2 Department of Psychiatry, University of British Columbia, Vancouver, Canada

* olisalotenna@gmail.com

## Abstract

Mental healthcare utilization in the US remains low, with persistent disparities observed across population groups. However, little is known about how sex and race/ethnicity jointly shape access to care. Intersectionality theory highlights the need to examine these dimensions together, as their combined influence may produce unique disadvantages not captured in single-axis analyses. This study utilized data from the 2009–2018 cycles of the National Health and Nutrition Examination Survey (NHANES). The relative differences (prevalence ratios) of mental healthcare utilization across intersecting sex and racial/ethnic groups were estimated using design-based log-binomial models. The absolute measure (prevalence differences) across these intersectional groups were obtained using linear probability regression models. Stratified analyses were conducted to examine how socioeconomic and need-related factors modified disparities. Overall, 9.1% of adults reported accessing mental health services in the preceding year. Marked disparities were observed across the intersectional groups. Hispanic males had the lowest utilization rates compared to Non-Hispanic (NH) White males, with an adjusted prevalence ratio (aPR) of 0.59 [95% CI: 0.47–0.73]. Among females, all minority racial/ethnic groups reported lower utilization compared with NH White females with aPRs ranging from 0.73 to 0.81. Within racial/ethnic groups, women generally accessed care more than men, though the magnitude of sex differences varied. Stratified analyses showed that disparities were magnified among those without insurance and attenuated at higher income levels. These results show that sex and race/ethnicity jointly shape patterns of mental healthcare utilization in the United States, producing compounded disadvantages for specific groups such as Hispanic men. Stratified analyses suggest that socio-economic status may modify these disparities, pointing to the role played by systemic inequities. These findings underscore the importance of intersectional approaches in population mental health research and policy. Future research should consider additional intersecting identities including sexual orientation and disability.

**Data availability statement:** The National Health and Nutrition Examination Survey (NHANES) dataset used for this study is publicly available from the National Center for Health Statistics repository on the Centre for Disease Control (CDC) website: https://www.cdc.gov/nchs/nhanes/index.html.

**Funding:** The authors received no specific funding for this work.

**Competing interests:** The authors have declared that no competing interests exist.

## Introduction

Mental health disorders affect approximately 1 in 8 people globally, with less than 75% of those affected receiving adequate care [1–3]. In the United States, the number of adults with any mental illness increased by 30% from 2008 to 2019, with over 50 million adults currently affected [4,5] Yet, more than half of them do not receive treatment [6,7]. Preventive mental health services like screening and early intervention are also critically lacking, especially for at-risk groups like individuals with chronic physical conditions [8]. Closing this persistent gap and doing so equitably represents an urgent public health priority.

Sociodemographic characteristics such as gender, race/ethnicity, sexual orientation and socioeconomic status are associated with mental health and access to care [9–11]. Whereas sociodemographic characteristics describe population groups and allow disparities to be quantified, social determinants of health (SDOH) represent the underlying societal conditions such as income, education, housing, and structural discrimination that generate inequities in the first place. The unequal distribution of these determinants helps explains why some populations experience greater barriers to mental healthcare and poorer outcomes [9,12]. Gender and race/ethnicity, in particular, are recognized as social constructs shaped by historical, cultural and political factors rather than biological differences. These categories often reflect unequal access to resources and opportunities, which in turn shape healthcare utilization and outcomes [13]. Although disparities by single-factor such as gender and race/ethnicity have been observed, how these factors intersect to influence mental healthcare utilization remains underexplored [14–19].

Intersectionality, a framework born out of Black women's concurrent experiences of racism and sexism emphasizes the interconnected nature of social inequalities [13,20,21]. The rise of intersectionality in health equity research underscores the limitations of single-factor approaches which treat social categories as independent, and thus fail to capture how the complex interplay of factors shape disparities [20,22]. Although quantitative methods are crucial for identifying health disparities at the population level, intersectional research has remained largely qualitative due to challenges in translating theoretical constructs into statistical models [19,20]. This study addresses this gap in relation to mental healthcare utilization by operationalising intersectionality using regression models with interaction terms, as proposed by Bauer [20]. Specifically, it aimed to investigate disparities in mental healthcare utilization among U.S adults across intersections of sex and race/ethnicity. While two axes of identity are not exhaustive, they represent critical starting points for operationalizing intersectionality in healthcare access research [20]. Identifying disproportionately affected intersectional groups can inform targeted interventions and health equity strategies.

## Methods

### Ethics statement

The National Health and Nutrition Examination Survey (NHANES) receives ethical approval from the CDC Institutional Review Board and all participants gave informed

consent [23]. This NHANES dataset is anonymous and publicly available, hence the study is exempt from institutional ethic review as outlined in the institution's and Tri-Council Policy Statements (TCPS2) on Ethical Conduct for Research Involving Humans [24,25]. We assessed the survey data from 3rd November 2024 and did not have access to any information that could identify participants at any stage.

## Data source

The study utilized aggregated data from five continuous cycles of the National Health and Nutrition Examination Survey (NHANES) from 2009-2018. The NHANES is an annual population-based, cross-sectional survey of civilian, non-institutionalized U.S. residents conducted by the Centers for Disease Control and Prevention (CDC) [23]. The survey collects sociodemographic, dietary, and health-related information from a nationally representative sample by employing a multi-stage probability sampling methodology. Detailed information on design, methodology and weighting are published on the CDC website.

## Study sample

Aggregated data from the five NHANES cycles included 49,693 participants, of which only adults aged 18 years and over were included in the study (n = 30,352). Following the exclusion of individuals with missing values in exposure and outcome, the final analytical sample was 30,340. Fig 1 illustrates the process of creating the analytical sample.

## Outcome

The outcome, 'mental healthcare utilization' reflected whether respondents accessed mental health services in the past twelve months and was ascertained by an affirmative response to the question: "*During the last 12 months, have you seen or talked to a mental health professional about your health?*" [26].

## Exposure

The exposure variables were sex and race/ethnicity as collected by NHANES. Sex at birth was categorized as male or female, and participants were grouped by race/ethnicity variable (RIDRETH1) into four categories (Non-Hispanic White,

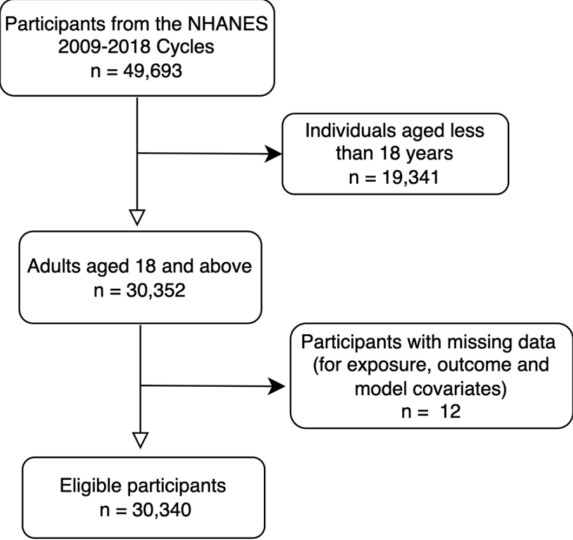

**Fig 1. Flowchart illustrating the sample selection process of adults aged 18+ from the NHANES cycles.**

Hispanic, Non-Hispanic Black and Non-Hispanic Other Race). The 'Non-Hispanic Other Race' category included individuals who identified as Non-Hispanic Asian, American Indian or Alaska Native, Native Hawaiian or Other Pacific Islander, and multi-racial individuals. This grouping reflects NHANES' racial and ethnic classifications and was used due to sample size limitations for more granular subgroup analysis.

While race, ethnicity, and sex are immutable characteristics, they serve as proxies for social experiences shaped by systemic factors such as structural racism, discrimination, and privilege. These variables were included to assess disparities in mental healthcare utilization, reflecting the influence of systemic and structural factors.

## Covariate selection

A directed acyclic graph (DAG) [*Dagitty* (version 3.1)] [27] was used to illustrate covariate relationship and select the final adjustment set [28]. Based upon the literature, the following factors: age, education level, marital status, income level, health insurance, perceived health status, depression and chronic physical condition were considered [Fig 2] [10,11,14,29–31]. Income level (low, middle, high) was categorized using recommended thresholds of poverty income ratio (PIR) [32]. Perceived health status refers to how participants rated their current general health status (NHANES variable HSD010) [23]. Depression was categorized by severity (none, mild, moderate, moderately severe and severe) based on the Patient Health Questionnaire-9 (PHQ-9) scores [33]. Chronic condition indicated whether participants had diabetes, hypertension, cancer or chronic kidney disease. Statistical significance of baseline differences was not used to

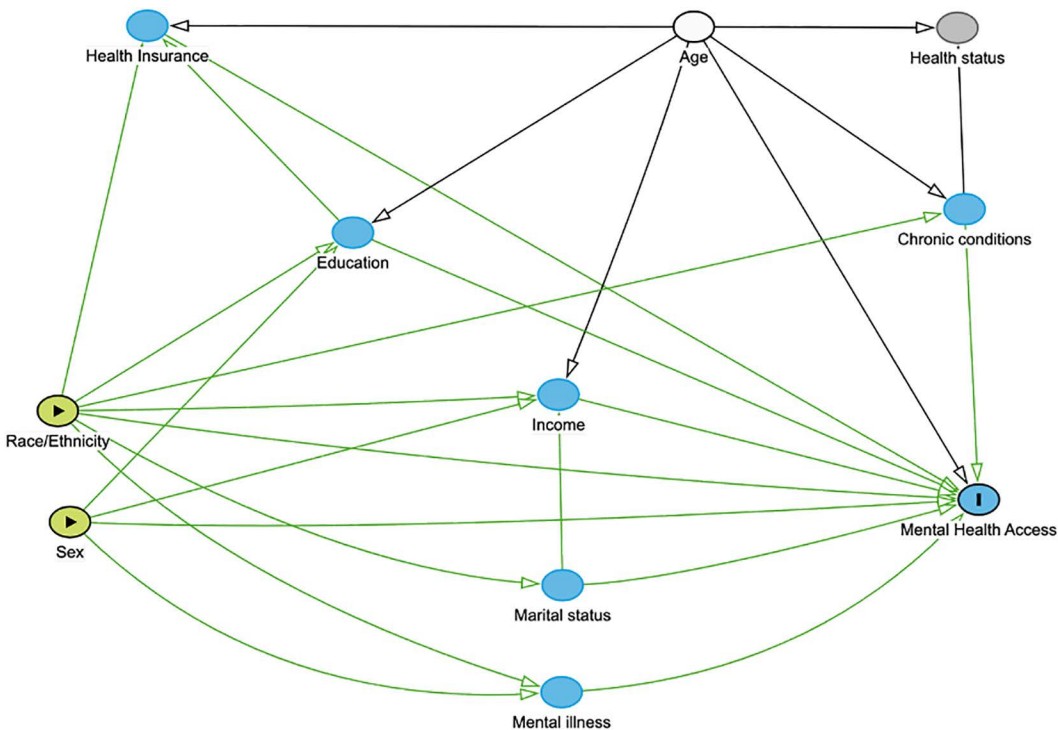

**Fig 2. Directed acyclic graph of the relationships between sex, race/ethnicity, mental health service utilization and associated factors.** *DAG Description:* Sex and race/ethnicity are represented as exposures. Mental health access is represented as the outcome. Arrows represent assumed causal relations informed by prior literature. Mediators (solid blue) include education, income, health insurance coverage, marital status, mental illness, and chronic conditions. They are represented as downstream consequences of sex and race/ethnicity that may affect access to mental healthcare. Health status (shaded grey) is considered a potential collider influenced by multiple variables. Thus, conditioning on it could introduce bias by opening backdoor biasing paths. Age (solid white) is an independent risk factor for the outcome which does not lie on the causal pathway.

select covariates, but confounder selection followed a more robust strategy using the DAG. Hence, the final adjustment set included independent risk factors for the outcome not positioned along the exposure-outcome causal pathway [28]. **Final Adjustment Set**: As shown in the DAG, age was the sole variable that did not introduce bias when adjusted for [Fig 2]. Since the outcome of interest is the total disparity in mental healthcare utilization associated with sex and race/ethnicity, adjustment for downstream variables such as education, income, and health insurance were avoided. These variables plausibly represent socio-structural consequences of sex and race/ethnicity via structural racism, discrimination, and gendered socialization. Hence, adjusting for them would block causal pathways of interest and bias the total association toward the null (overadjustment) [28]. Therefore, only age which does not lie on the exposure-outcome pathway was adjusted for as a precision variable.

## Statistical analysis

Descriptive statistics were conducted to summarize the study sample, with sociodemographic and health-related characteristics stratified by sex and race/ethnicity. Categorical variables were presented as counts with their respective weighted percentages. All analyses incorporated survey weights, clusters, and strata to account for the complex survey design [34].

## Primary analysis

To operationalize intersectionality in this quantitative analysis, interaction terms were used between sex and race/ethnicity in the regression models. To examine whether utilization rates differed across intersectional groups, design-based log-binomial and linear probability models estimated interactions on the multiplicative and additive scales respectively [20,35]. Prevalence ratios (PR) and Prevalence Differences (PD), with their 95% confidence intervals (95% CI) and significance levels, were obtained via simple slope/effects analysis. Overall missingness and missingness within variables in the model were less than 5% respectively, hence a complete case analysis was conducted [36].

## Stratified/Sensitivity analyses

To assess the consistency of the primary results as well as examine the role of socioeconomic and need-related factors in shaping the observed associations, stratified analyses were conducted by key variables: education level (high school and below vs. college and above), income level (low vs. middle vs. high), and health insurance status (insured vs. uninsured). Stratified analysis was also done by mental illness (depression – yes/no) to assess whether the observed results differed by mental health need. Depression severity (PHQ-9) was used as a proxy for clinical need, given the absence of validated measures for other mental disorders which precludes stratification for other conditions. Within each stratum, separate design-based log-binomial regression models including the interaction term between sex and race/ethnicity were fitted. The stratified analyses probed heterogeneity in the sex×race/ethnicity association without conditioning away the pathways that constitute the disparity itself. This approach allowed for an extension of the intersectional analysis beyond race/ethnicity and sex by evaluating whether the disparities observed in the primary analysis are modified by stratification variables.

To address missingness in key stratification variables, such as income level (10.5%) and depression (14.2%), we conducted multiple imputation. Little's MCAR test (p < 0.001) and bivariate analyses respectively revealed that missingness was not completely at random (MCAR) but could be explained by observed data. Under the missing at random (MAR) assumption, our imputation model included variables from the primary model, those requiring imputation (income level, education level, health insurance, and depression), and auxiliary variables (annual family income, marital status, chronic condition, health status, and survey strata). Five imputations with 20 iterations each were performed using logistic regression for binary variables and polynomial regression for categorical variables. The number of iterations was determined by monitoring the trace plots until they demonstrated convergence. Individuals with missing outcome data (n = 12) were excluded, resulting in a final analytical sample of 30,340. Design-adjusted multivariable regression models were run on

each imputed dataset, and results were pooled using Rubin's rule which combines the within- and between- imputation variances [37].

All statistical analyses were conducted using R (v4.3.1). The *survey* package was used to fit the design-adjusted models [38], and *mice* package was used for missing data analysis. This study followed the Strengthening the Reporting of Observational Studies in Epidemiology (STROBE) reporting guidelines as well as widely adopted recommendations for reporting on sex/gender and race/ethncity [39–41].

## Results

### Sample characteristics

The study sample consisted of 30,340 participants, with a nearly even distribution of males (48.5%) and females (51.5%). Tables 1 and 2 present the characteristics of the participants stratified by sex and race/ethnicity respectively. The counts were from the eligible sample while the percentages were derived using survey features, hence represent the national distribution. Significant sex and race-ethnic differences were observed across most demographic and health-related characteristics including education, income level, marital status and health insurance. The sample was predominantly Non-Hispanic (NH) White (64.8%), with small but significant differences in racial/ethnic composition between sexes. Overall, 9.1% of the U.S population accessed mental health services. Females reported a significantly higher proportion of mental health service utilization than males (9.7% vs 8.4%). Stratification by race/ethnicity revealed distinct differences in sociodemographic characteristics and health outcomes. For example, Hispanics had the highest proportion of low-income individuals (30.9%), and NH Whites were significantly more likely to report *excellent* and *very good* health compared to other race/ethnic groups. Mental healthcare utilization was highest among NH Whites (9.8%) and lowest among Hispanic and Other Race groups (7.3% each) [Table 2].

### Mental healthcare utilization across intersectional groups

Significant disparities were observed in both relative and absolute prevalence estimates of mental healthcare utilization across sex and race/ethnic intersectional groups [Table 3]. Compared to NH White males, Hispanic males were the least likely (41% less likely, aPR = 0.59 [0.47–0.73]) to utilize services, followed by males in the Other race group (33% less likely, aPR = 0.66 [0.50–0.88]). NH Black males were also less likely to utilize mental health services compared to NH White males. However, this association was insignificant (aPR = 0.91 [0.75–1.12]). Among females, Hispanic, NH Blacks and women in the Other race category were significantly less likely (25%, 19% and 27% less likely respectively) to utilize mental health services compared to NH White females.

Within the same race-ethnic group, females were more likely to utilize services than males. While NH White (aPR = 1.16 [1.00, 1.36]) and Hispanic females (aPR = 1.49 [1.22, 1.80]) had significantly higher probabilities of utilizing services than their male counterparts, this association was not significant among NH Blacks and individuals in the NH Other race category.

Results on the absolute/additive scale were consistent with the relative findings but indicated smaller differences [Table 3].

### Stratified analysis

The results of the stratified analyses were largely consistent with the primary analysis, although few subgroup differences emerged across education, income, health insurance, and depression status. [Table 4].

### Education

Compared to the primary analysis, stratification by education revealed notable differences among NH Black individuals. Among those with *high school education or less*, NH Black males had higher mental health service utilization than NH

**Table 1. Sample characteristics by sex among US adults in the 2009-2018 NHANES.**

| Characteristic (%) | Overall 30,340 | Male 14,707(48.5) | Female 15,633 (51.5) | p value |
|---|---|---|---|---|
| **Age** | | | | |
| 18-39 | 11,147 (38.4) | 5,418 (39.7) | 5,729 (37.2) | 0.017 |
| 40-59 | 9,442 (35.5) | 4,488 (35.9) | 4,954 (35.2) | |
| 60-79 | 7,810 (21.6) | 3,887 (20.8) | 3,923 (22.4) | |
| 80+ | 1,941 (4.4) | 914 (3.5) | 1,027 (5.3) | |
| **Ethnicity** | | | | |
| Non-Hispanic White | 11,699 (64.8) | 5,793 (65.2) | 5,906 (64.5) | 0.003 |
| Hispanic | 7,671 (15.0) | 3,598 (15.6) | 4,073 (14.5) | |
| Non-Hispanic Black | 6,611 (11.5) | 3,177 (10.7) | 3,434 (12.3) | |
| Other race* | 4,359 (8.6) | 2,139 (8.6) | 2,220 (8.7) | |
| **Marital status** | | | | |
| Married/with partner | 16,864 (62.5) | 8,881 (65.8) | 7,983 (59.5) | <0.001 |
| Divorced/Separated | 4,141 (12.7) | 1,697 (10.6) | 2,444 (14.7) | |
| Never married | 5,453 (18.8) | 2,769 (21.1) | 2,684 (16.8) | |
| Widowed | 2,345 (5.9) | 590 (2.5) | 1,755 (9.0) | |
| **Education Level** | | | | |
| High School or Less | 7,235 (15.6) | 3,672 (16.4) | 3,563 (14.9) | <0.001 |
| High School Graduate | 7,447 (24.1) | 3,780 (25.4) | 3,667 (22.9) | |
| Some College | 8,639 (30.8) | 3,843 (28.7) | 4,796 (32.7) | |
| College Graduate or Above | 6,857 (29.5) | 3,334 (29.4) | 3,523 (29.6) | |
| **Income** | | | | |
| Low Income | 6,413 (15.8) | 2,843 (14.2) | 3,570 (17.03) | <0.001 |
| Middle Income | 14,157 (48.6) | 6,937 (47.9) | 7,220 (49.1) | |
| High Income | 65,888 (35.6) | 3,400 (37.8) | 3,188 (33.6) | |
| **Mental Healthcare Utilization** | | | | |
| No | 27,753 (90.9) | 13,567 (91.6) | 14,186 (90.3) | <0.001 |
| Yes | 2,587 (9.1) | 1,140 (8.4) | 1,447 (9.7) | |
| **Depression** | | | | |
| None | 1,9471 (76.4) | 10,177 (80.8) | 9,294 (72.3) | <0.001 |
| Mild | 4,220 (15.5) | 1,772 (13.3) | 2,448 (17.7) | |
| Moderate | 1,443 (5.1) | 544 (3.7) | 899 (6.4) | |
| Severe | 904 (3.0) | 316 (2.2) | 588 (3.7) | |
| **Chronic Conditions** | | | | |
| No | 15,662 (58.5) | 7,607 (59.0) | 8,055 (58.1) | 0.485 |
| Yes | 13,194 (41.5) | 6,353 (41.0) | 6,841 (41.9) | |
| **Health status** | | | | |
| Excellent | 1,879 (8.7) | 1,031 (9.0) | 848 (8.3) | <0.001 |
| Very good | 6,830 (32.5) | 3,393 (31.3) | 3,437 (33.7) | |
| Good | 10,773 (40.5) | 5,326 (41.8) | 5,447 (39.2) | |
| Fair | 5,407 (15.7) | 2,569 (15.7) | 2,838 (15.8) | |
| Poor | 966 (2.6) | 396 (2.1) | 570 (3.0) | |
| **Health insurance** | | | | |
| No | 6,278 (17.2) | 3,372 (19.5) | 2,906 (15.0) | <0.001 |
| Yes | 24,004 (82.8) | 11,310 (80.5) | 12,694 (85.0) | |

**\*** *Other race* includes Non-Hispanic Asians, individuals of other racial/ethnic identity (e.g., American Indian) and multiracial individuals.

**Table 2.** Sample characteristics stratified by race-ethnicity among US adults aged 18 years or more in the 2009-2018 NHANES.

| Level (%) | Overall | Non-Hispanic White | Hispanic | Non-Hispanic Black | Other race | p-value |
|---|---|---|---|---|---|---|
| N | 3,0340 | 11,699(64.8) | 7,671(15.0) | 6,611(11.5) | 4,359 (8.6) | |
| **Sex** | | | | | | 0.003 |
| Male | 14,707 (48.2) | 5,793 (48.5) | 3,598 (50.0) | 3,177(44.7) | 2,139 (47.8) | |
| Female | 15,633 (51.8) | 5,906 (51.5) | 4,073 (50.0) | 3,434 (55.3) | 2,220 (52.2) | |
| **Age** | | | | | | <0.001 |
| 18-39 | 11,147 (38.4) | 3,886 (33.4) | 3,040 (52.6) | 2,367 (43.3) | 1,854 (44.6) | |
| 40-59 | 9,442 (35.5) | 3,382 (35.9) | 2,433 (33.4) | 2,132 (36.1) | 1,495 (35.4) | |
| 60-79 | 7,810 (21.6) | 3,065 (25.0) | 2,004 (12.5) | 1,867 (18.0) | 874 (17.4) | |
| 80+ | 1,941 (4.4) | 1,366 (5.7) | 194 (1.5) | 245 (2.6) | 136 (2.6) | |
| **Marital status** | | | | | | <0.001 |
| Divorced/Separated | 4,141 (12.7) | 1,607 (12.5) | 1,062 (12.8) | 1,141 (16.6) | 331 (9.0) | |
| Married/with partner | 16,864 (62.5) | 6,842 (65.2) | 4,550 (64.2) | 2,686 (42.6) | 2,786 (66.0) | |
| Never married | 5,453 (18.8) | 1,693 (15.7) | 1,096 (19.4) | 1,865 (34.2) | 799 (20.9) | |
| Widowed | 2,345 (5.9) | 1,139 (6.5) | 460 (3.6) | 551 (6.6) | 195 (4.1) | |
| **Education level** | | | | | | <0.001 |
| High School or Less | 7,235 (15.6) | 1,795 (10.1) | 3,384 (38.4) | 1,373 (18.8) | 683 (13.9) | |
| High School Graduate | 7,447 (24.1) | 2,999 (23.9) | 1,745 (24.8) | 1,898 (29.1) | 805 (18.1) | |
| Some College | 8,639 (30.8) | 3,754 (32.0) | 1,689 (24.7) | 2,195 (34.3) | 1,001 (27.1) | |
| College Graduate or above | 6,857 (29.5) | 3,121 (34.1) | 788 (12.1) | 1,093 (17.8) | 1,855 (40.9) | |
| **Income level** | | | | | | <0.001 |
| Low Income | 6,413 (15.8) | 1,902 (10.4) | 2,142 (30.9) | 1,647 (27.9) | 722 (16.6) | |
| Middle Income | 14,157 (48.6) | 5,692 (46.4) | 3,546 (54.2) | 3,072 (52.7) | 1,847 (50.2) | |
| High Income | 6,588 (35.6) | 3,329 (43.1) | 874 (14.8) | 1,112 (19.5) | 1,273 (33.2) | |
| **Mental Healthcare Utilization** | | | | | | <0.001 |
| No | 27,753 (90.9) | 10,522(90.2) | 7,123 (92.7) | 6,016 (91.1) | 4,092 (92.7) | |
| Yes | 2,587 (9.1) | 1,177 (9.8) | 548 (7.3) | 595 (8.9) | 267 (7.3) | |
| **Chronic conditions** | | | | | | <0.001 |
| No | 15,662 (58.5) | 5,790 (56.2) | 4,268 (68.7) | 2,989 (54.1) | 2,615 (64.5) | |
| Yes | 13,194 (41.5) | 5,499 (43.8) | 2,903 (31.3) | 3,286 (45.9) | 1,506 (35.5) | |
| **Depression** | | | | | | <0.001 |
| None | 19,471 (76.4) | 7,672 (77.1) | 4,806 (74.9) | 4,218 (73.7) | 2,775 (77.7) | |
| Mild | 4,220 (15.5) | 1,700 (15.3) | 1,906 (16.1) | 942 (17.0) | 482 (14.3) | |
| Moderate | 1,443 (5.1) | 594 (4.9) | 389 (5.3) | 315 (5.9) | 145 (5.5) | |
| Severe | 904 (3.0) | 368 (2.8) | 278 (3.6) | 189 (3.4) | 69 (2.6) | |
| **Health status** | | | | | | <0.001 |
| Excellent | 2,416 (10.5) | 1,059 (11.4) | 489 (8.1) | 462 (8.4) | 406 (10.9) | |
| Very good | 6,830 (31.9) | 3,447 (36.5) | 1,065 (18.7) | 1,303 (23.5) | 1,015 (29.4) | |
| Good | 10,773 (39.7) | 4,063 (38.2) | 2,743 (42.7) | 2,446 (42.5) | 1,521 (42.2) | |
| Fair | 5,407 (15.4) | 1,562 (11.9) | 2,016 (26.5) | 1,326 (22.3) | 503 (14.8) | |
| Poor | 966 (2.5) | 335 (2.0) | 341 (4.1) | 208 (3.3) | 82 (2.7) | |
| **Health insurance** | | | | | | <0.001 |
| No | 6,278 (17.2) | 1,638 (11.5) | 2,666 (38.4) | 1,265 (21.8) | 709 (16.8) | |
| Yes | 24,004 (82.8) | 10,049(88.5) | 4,986 (61.6) | 5,329 (78.2) | 3,640 (83.2) | |

**Table 3. Prevalence ratios (PR) and prevalence differences (PD) of mental healthcare utilization by sex and race/ethnicity (95% Confidence Intervals).**

| Race/Ethnicity PR & PD (95% CI) | Male | Female | Female vs Male (within strata of race-ethnicity) |
|---|---|---|---|
| **Relative Scale Unadjusted PR** | | | |
| Non-Hispanic White | Ref^ | Ref | 1.14 (0.97, 1.32) |
| Hispanic | 0.65 (0.52, 0.80)*** | 0.84 (0.70, 0.99)* | 1.46 (1.20, 1.79)*** |
| Non-Hispanic Black | 0.97 (0.80, 1.19) | 0.87 (0.73, 1.05) | 1.02 (0.87, 1.20) |
| Other Race | 0.70 (0.53, 0.92)** | 0.79 (0.61, 1.00)* | 1.27 (0.92, 1.75) |
| **Adjusted PR** | | | |
| Non-Hispanic White | Ref | Ref | 1.16 (1.00, 1.36)* |
| Hispanic | 0.59 (0.47, 0.73)*** | 0.75 (0.63, 0.90)*** | 1.49 (1.22, 1.80)** |
| Non-Hispanic Black | 0.91 (0.75, 1.12) | 0.81 (0.68, 0.98)* | 1.04 (0.89, 1.21) |
| Other Race | 0.66 (0.50, 0.88)** | 0.73 (0.57, 0.92)* | 1.27 (0.92, 1.77) |
| **Absolute Scale Unadjusted PD** | | | |
| Non-Hispanic White | Ref | Ref | 0.01 (-0.00, 0.03) |
| Hispanic | -0.03 (-0.05, -0.02)*** | -0.02 (-0.03, -0.00)* | 0.03 (0.01, 0.04)*** |
| Non-Hispanic Black | -0.00 (-0.02, 0.02) | -0.01 (-0.03, 0.00) | 0.00 (-0.01, 0.02) |
| Other Race | -0.03 (-0.05, -0.01)** | -0.02 (-0.04, -0.00)* | 0.02 (-0.01, 0.04) |
| **Adjusted PD** | | | |
| Non-Hispanic White | Ref | Ref | 0.01 (-0.00, 0.03) |
| Hispanic | -0.03 (-0.05, -0.01)*** | -0.02 (-0.03, -0.00)** | 0.03 (0.01, 0.04)*** |
| Non-Hispanic Black | -0.00 (-0.02, 0.01) | -0.02 (-0.03, 0.00) | -0.00 (-0.01, 0.01) |
| Other Race | -0.03 (-0.05, -0.01)** | -0.02 (-0.04, -0.00)* | 0.02 (-0.01, 0.04) |

Adjusted estimates are adjusted for Age.

^Ref are the reference groups of comparisons. Within male and female categories, individuals in other intersectional groups are compared to Non-Hispanic Whites.

*indicates statistical significance at p values: *<0.05, **<0.01 ***<0.001

White males (aPR = 1.26, 95% CI: 0.96–1.65), contrasting with the overall finding of lower utilization among NH Black males in the primary analysis. NH Black males with *high school education or less* were also more likely to utilize services compared to their female counterparts (NH Black females) [Table 4]. Although these differences were not statistically significant, they indicate a directional reversal in disparities among lower-educated NH Black individuals compared to results from the primary analysis. Conversely, among those with higher education (*college and above*), findings mirror the main analysis with NH Black males having significantly lower utilization than NH White males, and NH Black females having higher utilization than NH Black males.

### Income

Among individuals in the *middle-income* category, results from the stratified analysis aligned closely with the overall findings. However, among those with *low income*, NH Black females had lower utilization than NH Black males (aPR = 0.72 [0.52, 0.99]), again reversing the sex pattern from the primary analysis. This reversal is consistent with the pattern seen in the lower education stratum, suggesting a potential intersection of sex and socio-economic status in shaping service use. Among individuals with high-income, disparities across intersectional groups were not statistically significant. This contrasts with the significant disparities seen in the main analysis, suggesting that higher income may attenuate disparities.

**Table 4. Results of stratified analysis by education level, income level, health insurance and depression status showing adjusted prevalence ratios (PR) with 95% CIs.**

| Stratification Variable | Male aPR (95% CI) | Female aPR (95% CI) | Female vs Male aPR (95% CI) |
|---|---|---|---|
| **Education High School or below** | | | |
| Non-Hispanic White | Ref | Ref | 1.17 [0.91; 1.51] |
| Hispanic | 0.58 [0.44,0.75]*** | 0.74 [0.58,0.93]** | 1.51 [1.17,1.92]*** |
| Non-Hispanic Black | 1.26 [0.96, 1.65] | 0.93 [0.72, 1.21] | 0.87 [0.68; 1.12] |
| Other race | 0.89 [0.59, 1.35] | 0.94 [0.64, 1.38] | 1.25 [0.73, 2.12] |
| **College and above** | | | |
| Non-Hispanic White | Ref | Ref | 1.14[0.96, 1.35] |
| Hispanic | 0.73 [0.54,0.99]* | 0.88[0.69,1.12] | 1.38[1.02; 1.88]* |
| Non-Hispanic Black | 0.70 [0.55, 0.88]** | 0.78[0.63, 0.97]* | 1.28[0.99; 1.67] |
| Other race | 0.58[0.39, 0.86]** | 0.66 [0.50, 0.87]** | 1.31[0.83, 2.06] |
| **Income Low Income** | | | |
| Non-Hispanic White | Ref | Ref | 1.22 [0.99, 1.51] |
| Hispanic | 0.46 [0.36, 0.60]*** | 0.47[0.36, 0.62]*** | 1.01 [0.73, 1.39] |
| Non-Hispanic Black | 1.01 [0.77, 1.31] | 0.67 [0.50, 0.99]** | 0.72 [0.52, 0.99] |
| Other race | 0.68 [0.43, 1.10] | 0.62 [0.44, 0.90]** | 1.06 [0.64, 1.75] |
| **Middle Income** | | | |
| Non-Hispanic White | Ref | Ref | 1.15 [0.89, 1.49] |
| Hispanic | 0.56 [0.40, 0.79]*** | 0.75[0.46, 1.06] | 1.62 [1.20, 2.18]** |
| Non-Hispanic Black | 0.75 [0.55, 1.03] | 0.75 [0.52, 1.57] | 1.26 [0.93, 1.72] |
| Other race | 0.45[0.28, 0.75]*** | 0.57 [0.29, 1.14] | 1.65 [0.95, 2.82] |
| **High Income** | | | |
| Non-Hispanic White | Ref | Ref | 1.11 [0.84, 0.46] |
| Hispanic | 0.73 [0.46, 1.15] | 1.00[0.63, 0.90] | 1.54 [0.89, 2.67] |
| Non-Hispanic Black | 0.72 [0.49, 1.06] | 0.95 [0.57, 1.57] | 0.95 [0.57, 1.57] |
| Other race | 0.76 [0.43, 1.36] | 0.89 [0.44, 1.73] | 0.87 [0.44, 1.73] |
| **Insurance Uninsured** | | | |
| Non-Hispanic White | Ref | Ref | 1.75[1.21; 2.53]** |
| Hispanic | 0.52 [0.34,0.81]** | 0.30 [0.21,0.41]*** | 1.00 [0.73,1.37] |
| Non-Hispanic Black | 0.87[0.54, 1.37] | 0.45 [0.30,0.70]*** | 0.92[0.59; 1.43] |
| Other race | 0.73 [0.33, 1.62] | 0.43 [0.25, 0.76]** | 1.04[0.47, 2.32] |
| **Insured** | | | |
| Non-Hispanic White | Ref | Ref | 1.11[0.94, 1.28] |
| Hispanic | 0.73 [0.57,0.94]* | 1.00[0.84,1.20] | 1.51[1.20; 1.92]*** |
| Non-Hispanic Black | 0.99[0.80, 1.12] | 0.90[0.74, 1.08] | 1.00[0.86, 1.17] |
| Other race | 0.67[0.49, 0.90]** | 0.78[0.79, 1.01] | 1.28[0.89, 1.84] |
| **Depression status No Depression** | | | |
| Non-Hispanic White | Ref | Ref | 1.08[0.86, 1.37] |
| Hispanic | 0.55[0.42,0.72]*** | 0.75[0.57,0.98]* | 1.48[1.13,1.91]** |
| Non-Hispanic Black | 0.88[0.66, 1.17] | 0.73[0.52, 0.90]** | 0.84[0.66; 1.09] |
| Other race | 0.57 [0.34, 0.93]* | 0.73 [0.52, 1.04] | 1.40[0.80, 2.46] |
| **Depression** | | | |
| Non-Hispanic White | Ref | Ref | 0.99[0.81, 1.21] |
| Hispanic | 0.61[0.46,0.81]*** | 0.70[0.57,0.87]*** | 1.14[0.89, 1.46] |
| Non-Hispanic Black | 0.90[0.70, 1.15] | 0.84[0.68, 1.03] | 0.92[0.76, 1.12] |
| Other race | 0.82[0.57, 1.17] | 0.76 [0.57, 1.01] | 0.91[0.62, 1.37] |

*indicates statistical significance at p values: *<0.05, **<0.01 ***<0.001

### Insurance

Stratified results by insurance status were broadly consistent with the primary analysis, particularly in terms of direction. However, disparities were more pronounced among uninsured females, with ethnic minority females having even lower utilization rates than in the main analysis. For example, uninsured Hispanic women had 70% lower utilization than uninsured NH White women (aPR = 0.30, 95% CI: 0.21–0.41), a sharper disparity than observed in the primary analysis (25% lower utilization, aPR = 0.75), indicating that lack of insurance may exacerbate disparities among women.

### Depression

The adjusted prevalence ratios (aPRs) for mental healthcare utilization stratified by depression status were largely consistent with the primary analysis suggesting that depression status did not substantially alter the observed disparities.

## Discussion

### Primary findings

This study examined disparities in mental healthcare utilization among U.S adults across intersections of sex and race-ethnicity using nationally representative NHANES data. Only 9.1% of adults reported using mental health services in the prior year, underscoring the persistence of the treatment gap despite high levels of need [6,7]. The intersectional analysis revealed that Hispanic males had the lowest utilization rates, being 41% less like than NH White males to access care. Among females, all minority racial/ethnic groups had significantly lower utilization than NH White females. Within racial/ethnic groups, females generally accessed services more than males, although the magnitude of the sex gap differed across groups. Absolute prevalence differences mirrored relative measures but were smaller, reflecting the overall low baseline utilization rate.

These findings extend prior single-axis analyses by demonstrating how sex and race/ethnicity jointly shape access to care. For instance, the varying magnitude of sex gaps across racial/ethnic groups—from 49% higher rates for Hispanic females to only 4% higher for NH Black females relative to males—illustrates how sex and race/ethnicity interact differently across groups. Consistent with intersectionality theory, the disparities observed cannot be reduced to either race/ethnicity or sex alone but emerge from their combined influence [13,16,22].

The persistence of disparities even when stratified by depression status suggests that differences in need may not fully explain unequal utilization. This aligns with evidence showing that despite having comparable or higher prevalence of mental disorders, Hispanic and NH Blacks access care at lower rates than NH Whites [42,43]. Thus, intersecting social norms, systemic inequities, and structural barriers rather than need alone shape access among certain intersectional groups. For example, Hispanic men's especially low utilization may reflect the interplay of traditional masculine norms that discourage help-seeking, alongside structural disadvantages including lower insurance coverage, language barriers, and immigration-related challenges [44–49]. This reinforces Crenshaw's intersectionality framework and broader evidence that illustrate how healthcare access is shaped not only by clinical need but also by overlapping effects of social identities and systemic inequities [17,21,31].

### Stratified analyses

Stratified analyses highlighted how socioeconomic status modifies observed disparities. The amplification of disparities among uninsured and low-income groups especially Hispanic and NH Blacks suggests that insurance coverage and income level are important structural factors. Furthermore, the absence of significant differences in utilization among high-income individuals suggests that socioeconomic advantages may attenuate disparities. Education also appeared to shape patterns among NH Blacks: Black men with lower education levels showed unexpectedly higher utilization relative to NH White men, while NH Black women in the same group reported lower utilization than NH Black men. This

reversal may reflect pathways into treatment through coercive or involuntary routes, such as criminal justice referrals, which disproportionately affect lower-educated NH Black men [50,51]. Together, the primary and stratified study results demonstrate the utility of intersectional analyses for revealing heterogeneity within groups that single-factor approaches obscure.

### Policy and practice implications

The study findings underscore the need for health policies and interventions that explicitly recognize intersecting inequities in access. The patterns observed suggest that culturally responsive and gender-sensitive strategies should be tested for effectiveness among key groups such as Hispanic men, where disparities are most pronounced. Promising approaches include the integration of mental health services into trusted community and faith-based organizations and the development of bilingual and culturally-competent outreach programs that challenge traditional masculine norms around help-seeking [29,49,52]. At the structural level, reforms that expand insurance coverage, reduce cost barriers, and improve geographic accessibility are likely to benefit underserved groups [4,52,53]. More broadly, policies addressing social determinants of health including education and employment are also essential to reducing disparities in mental healthcare utilization [52,54]. Intersectionality provides a framework for targeting interventions to the groups that are most disadvantaged at the overlap of multiple social identities.

### Limitations

Several limitations should be considered when interpreting these findings. First, NHANES collects binary sex assigned at birth but does not capture gender identity [26]. While sex and gender are correlated, they are distinct constructs. Hence, the binary sex variable does not reflect the social, cultural, and individual experiences of gender identity and also limits the examination of disparities in gender-diverse populations. Likewise, the use of broad and heterogenous racial/ethnic categories such as "Non-Hispanic Other' obscures important within-group differences including among equity-deserving groups like indigenous populations.

Second, sexual orientation – an important factor in intersectional analysis – could not be included in the analyses due to methodological challenges in the NHANES dataset including insufficient sample sizes for non-heterosexual groups and high rates of non-specific responses [23]. This highlights the need for improved data collection methods that better capture sexual orientation while maintaining statistical power for meaningful analysis. Furthermore, intersectionality is not limited to gender and race/ethnicity. Although the stratified analysis enabled consideration of other axes such as socioeconomic status, important factors like disability remain unexplored and can be the focus of future research.

Third, while stratification by depression severity provided useful insight, NHANES' lack of consistently collected validated measures for other mental disorders restricted the scope of the need-based analyses. Fourth, the reliance on self-reported service utilization may be subject to recall and social desirability bias. Finally, these findings may not be generalizable outside the United States because while intersectionality is broadly applicable framework, the drivers of inequity differ across contexts. Despite these limitations, the large, nationally representative sample and use of intersectional analytic methods provide a robust contribution to understanding inequities in mental-healthcare utilization.

### Conclusion

This study demonstrated significant intersectional disparities in mental health service utilization in the United States. Hispanic men and women from racial/ethnic minority groups had comparatively lower utilization rates, with socioeconomic position further shaping access. These results underscore the need to consider multiple, intersecting identities when designing policies and interventions to improve mental healthcare access.

## Author contributions

**Conceptualization:** Lotenna Olisaeloka.

**Data curation:** Lotenna Olisaeloka, Gentille Musengimana.

**Formal analysis:** Lotenna Olisaeloka, Esteban J. Valencia.

**Investigation:** Lotenna Olisaeloka.

**Methodology:** Lotenna Olisaeloka, Esteban J. Valencia, Ehsan Karim.

**Project administration:** Lotenna Olisaeloka.

**Resources:** Lotenna Olisaeloka.

**Supervision:** Daniel Vigo, Ehsan Karim.

**Writing – original draft:** Lotenna Olisaeloka, Gentille Musengimana.

**Writing – review & editing:** Lotenna Olisaeloka, Gentille Musengimana, Esteban J. Valencia, Daniel Vigo, Ehsan Karim.

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
