## [Decision Letter · Decision Letter 0]

6 Jul 2025

PGPH-D-25-01113

Intersectional Disparities in Mental Healthcare Utilization by Sex and Race/Ethnicity among US Adults: An NHANES Study

Dear Dr. Olisaeloka,

Thank you for submitting your manuscript to PLOS Global Public Health. After careful consideration, we feel that it has merit but does not fully meet PLOS Global Public Health’s publication criteria as it currently stands. Therefore, we invite you to submit a revised version of the manuscript that addresses the points raised during the review process.

We look forward to receiving your revised manuscript.

Kind regards,

Susan Julia Chand, PhD

Guest Editor

Journal Requirements:

Additional Editor Comments (if provided):

Reviewers' comments:

Reviewer's Responses to Questions

**Comments to the Author**

1. Does this manuscript meet PLOS Global Public Health’s publication criteria?

Reviewer #1: Yes

Reviewer #2: Yes

Reviewer #3: Partly

Reviewer #4: Partly

2. Has the statistical analysis been performed appropriately and rigorously?

Reviewer #1: Yes

Reviewer #2: I don't know

Reviewer #3: No

Reviewer #4: Yes

3. Have the authors made all data underlying the findings in their manuscript fully available (please refer to the Data Availability Statement at the start of the manuscript PDF file)?

Reviewer #1: Yes

Reviewer #2: Yes

Reviewer #3: Yes

Reviewer #4: Yes

4. Is the manuscript presented in an intelligible fashion and written in standard English?

Reviewer #1: Yes

Reviewer #2: Yes

Reviewer #3: Yes

Reviewer #4: Yes

Reviewer #1: This is a well-conceived and competently executed manuscript that applies intersectional theory to a vital public health issue: disparities in access to mental healthcare.

Key strengths include:

Use of a nationally representative dataset (NHANES, 2009–2018) and transparent methodology.

Operationalisation of intersectionality using interaction terms and both multiplicative and additive modelling.

Meaningful stratified analysis by socioeconomic and health-related variables (education, income, insurance, depression), which adds important nuance.

Clear policy implications, including recommendations for culturally responsive services and structural reform.

A few suggestions for improvement:

Clarify and define race/ethnicity categories early in the text, and consider commenting on the limitations of these broad categories (especially the heterogeneous “Other” group).

Provide greater clarity on limitations of sex vs. gender measures, and the consequences for interpretability, especially in light of intersectionality theory.

Strengthen the policy/practice recommendations by providing examples of successful interventions or promising approaches for specific subgroups (e.g., Hispanic men).

Conduct a final editorial proofread to catch minor inconsistencies and improve readability.

Reviewer #2: 1. Using personal pronouns (like "I" or "we") in the abstract of a research article is not recommended, especially in formal, academic, or scientific writing.

Passive voice or third-person constructions are often used. e.g Instead of “We analyzed the data,” write “The data were analyzed.”

2. Two different references style used - please stick to one

3. Please avoid personal pronouns, this is an academic article. its not standard practice

4. Passive voice or third-person constructions should be used

5. please include a table footer to show the significance level. e.g p<0.05.

Also an asterisk sign beside a significant value

6. Please do not use and cite publications over 10 years

Reviewer #3: Thank you for the opportunity to review this paper. This is an important topic and a useful way to look at the intersectionality of mental health services utilization. Here are my questions / comments on the paper:

1) Why was the adjusted model limited to age? It says that the other covariates introduced bias when they were adjusted for - but that is not well explained. Furthermore, this is a paper about intersectionality, but excluding overlapping variabilities without more sufficient explanation seems like a mistake both statistically and theoretically. Furthermore, the paper goes on to do sensitivity analyses, but again considers them each individually, which is a weakness of this paper, particularly in light of its focus on intersectionality.

2) Why is depression the only condition included in the analysis? It may be the only one with severity, but including other conditions would strengthen this. Or to be clear that the topic is depression - not mental health conditions more broadly.

3) The first sentence of the introduction could be stronger. Did the prevalence change 30 years ago? Is this time period relevant for treatment efforts?

4) The DAG requires more explanation - what do the lines mean and where is that data coming from?

5) The Table 1 data is quite extensive and probably too much. I would consider reorganizing or presenting it more clearly. Also, with the p values nearly all below 0.05, this is further evidence for including covariates in the analyses.

6) The discussion needs a lot of work. Both to community the findings more clearly, but also the related literature is not well organized and sometimes problematic. The results should be presented similarly (eg, 41% less likely vs 0.75-0.81), but there is far too much of restating the results and the linkages to related literature is weak. For example, the paragraph - "several factors might contribute to these observed.... has the quality of care from providers, stigma, and structural barriers. These are massive topics and then to jump to potential explanations of why Hispanic males might have lower access tried to cover too much space. I would recommend revising.

3) Building on the challenges of the discussion, the policy and practice implications lack focus and aren't meaningfully related to the findings of this paper.

7) Limitations: This is the first mention of LGBTQ+ people, so it seems a bit too much on this and does not reflect the other limitations of this paper (including that intersectionality is more than race and gender. Or that race is a social construct. Using a huge data set. Lack of data on other mental health conditions. Etc.).

Reviewer #4: Thank you for the opportunity to review this very informative and engaging article.

General overview:

The interaction of socio-demographic factors, especially age and socioeconomic status, bidirectionally (positively or negatively) influences MH service utilization not only in this study setting. This article provides a comprehensive overview of key background factors influencing mental health utilization worldwide. Inadvertently, the help-seeking behaviors, individualized multifactorial interplay indicated by these socio-demographics, and relevant socio-cultural-environmental, and systemic inequalities are key salient differences and factors in Mental health utilization. The evidence from this study on the intersectionality of socio-demographics and mental health service utilization is crucial for understanding the causes for low uptake of available health services in the US.

However, ” culturally informed strategies and wider structural interventions...(Abstract & conclusion)” is beautiful but outside of data. What informs this conclusion? It seems MH services are available, but the actual uptake is affected by the sociodemographic characteristics. Also, there could be other systemic factors, like a lack of task-shifted services, that are affecting access and utilization of the services, not necessarily culturally informed strategies.

‘Social determinants of health’, a close concept to socio-demographics, is mentioned in the introduction and discussion. Adding a sentence or two explaining distinct causative and impactful differences in mental health utilization would help provide more clarity in their contributions.

Recommendation:

Consider the ‘culturally informed strategies and structural interventions’ as a recommendation for further study.

Limitation:

Please add why this article can not be generalized to the LMICs, or provide a brief relevance of this article to the LMICs in the discussion

**Do you want your identity to be public for this peer review?** For information about this choice, including consent withdrawal, please see our Privacy Policy

Reviewer #1: **Yes: ** Simon Browes

Reviewer #2: No

Reviewer #3: No

Reviewer #4: **Yes: ** Josephine Akellot

---

## [Editor Report · Decision Letter 1]

22 Sep 2025

Intersectional Disparities in Mental Healthcare Utilization by Sex and Race/Ethnicity among US Adults: An NHANES Study

PGPH-D-25-01113R1

Dear Dr Olisaeloka,

We are pleased to inform you that your manuscript 'Intersectional Disparities in Mental Healthcare Utilization by Sex and Race/Ethnicity among US Adults: An NHANES Study' has been provisionally accepted for publication in PLOS Global Public Health.

Best regards,

Susan Julia Chand, PhD

Guest Editor